# Long Non-Coding RNA *SNHG4* Expression in Women with Endometriosis: A Pilot Study

**DOI:** 10.3390/genes14010152

**Published:** 2023-01-05

**Authors:** Tomasz Szaflik, Hanna Romanowicz, Krzysztof Szyłło, Beata Smolarz

**Affiliations:** 1Department of Gynaecology, Oncological Gynaecology and Endometriosis Treatment, Polish Mother’s Memorial Hospital Research Institute, 93-338 Lodz, Poland; 2Laboratory of Cancer Genetics, Department of Pathology, Polish Mother’s Memorial Hospital Research Institute, Rzgowska 281/289, 93-338 Lodz, Poland

**Keywords:** lncRNA, SNHG4, endometriosis

## Abstract

Background: Endometriosis is a chronic disease of the genital organs that mainly affects women of reproductive age. The analysis of long non-coding RNA (lncRNA) in endometriosis is a novel field of science. Recently, attention has been drawn to *SNHG4*, which is incorrectly expressed in various human diseases, including endometriosis. Aim: The aim of this pilot study was to analyze the expression of lncRNA small nucleolar RNA host gene 4 (*SNHG4*) and to investigate its significance in endometriosis. Material and methods: LncRNA *SNHG4* expression was investigated in paraffin blocks in endometriosis patients (n = 100) and in endometriosis-free controls (n = 100) using a real-time PCR assay. Results: This study revealed a higher expression of *SNHG4* in endometriosis patients than in controls. A statistically significant relationship between expression level and *SNHG4* was found in relation to The Revised American Society for Reproductive Medicine classification of endometriosis, 1996, in the group of patients with endometriosis. Conclusion: This pilot study has revealed that gene expression in *SNHG4* plays an important role in the pathogenesis of endometriosis.

## 1. Introduction

Endometriosis is considered a debilitating gynecological pathology. This disease affects 10–15% of women of reproductive age and 35–50% of women with chronic pelvic pain and/or infertility [1]. Literature data indicate that the incidence of postmenopausal endometriosis is of approximately 2–5% [2,3]. A retrospective epidemiological study conducted on 42,079 women admitted to surgery with histologically confirmed endometriosis showed that 33,814 patients (80.36%) were in the premenopausal group (0–45 years), 7191 (17.09%) were in perimenopausal (45–55 years), and 1074 patients (2.55%) were in the postmenopausal group [4]

For endometriosis, the presence of endometrial cells outside the uterine cavity is characteristic. These cells are still active secretory and, reacting to hormonal changes occurring in the menstrual cycle, cause a chronic inflammatory response in the body [3,5,6]. Due to the non-specific symptoms of this disease, which can mimic those associated with pelvic inflammation or other conditions associated with chronic pelvic pain, the gold standard of the final diagnosis is surgery, followed by histopathological examination. This may explain the significant delay in the diagnosis of endometriosis, leading to late proper treatment. So far, it has not been possible to establish reliable laboratory biomarkers for this gynecological pathology [1]. The pathophysiology of this disease is not exactly understood. Among the etiological factors, congenital, environmental, epigenetic, autoimmune, and allergic factors are listed.

The main etiopathogenetic factor of endometriosis is still considered to be Sampson’s theory of retrograde menstruation. However, about 90% of unaffected women experience retrograde menstruation [1]. New evidence supports the hypothesis that the development of endometriosis is caused by the appearance of primitive endometrial cells outside the uterus during organogenesis. Any dysregulation of genes involved in the differentiation of the anatomical structures of the genitourinary system leads to various anomalies and can cause abnormal location of stem cells, which, combined with immune changes and the pro-inflammatory environment of the peritoneum, will determine the progression towards endometriosis [7].

Recently, in the context of endometriosis, long non-coding RNAs (lncRNAs) that cover more than 200 bp and are a subtype of non-coding RNAs (ncRNAs) have become the focus of interest [8,9,10,11,12]. Recent studies have highlighted lncRNA *SNHG4* (small nucleolar RNA host gene 4), which may be important in the context of endometriosis [13].

SNHGs are small nucleolar RNA (snoRNA) host genes that are present in the nucleus and cytoplasm [14]. LncRNA *SNHG4* belongs to the *SNHG* family, which has 22 members, from *SNHG1* to *SNHG22* [15]. SNHGs play a significant role in cancer as well as in other human diseases. Research indicates that *SNHG3* is a new lncRNA oncogene. It is abnormally expressed in lung cancer, osteosarcoma, and hepatocellular carcinoma (HCC) [16]. Increased *SNHG3* expression is associated with the process of proliferation, migration, and invasion of cancer cells. Studies point to the role of SNHG1 as a useful biomarker for diagnosing, predicting, and treating cancer in humans [17,18]. *SNHG5*, *SNHG7*, *SNHG12*, and *SNHG16* are known to promote tumour progression [19,20,21,22]. SNHG4 expression has been analyzed in tumors [23,24,25,26,27,28,29] and endometriosis [27]. *SNHG4* has been shown to be expressed in the above diseases. This suggests that *SNHG4* may be an important factor in their formation and development. However, no association was demonstrated between *SNHG4* expression and clinical characteristics of patients. There was also no association with the prognosis of these diseases [23,24,25,26,27,28,29,30].

In vivo studies have shown that *SNHG4* may play a carcinogenic role. Knockdown SNHG4 may inhibit the growth of tumours including renal cell carcinoma (RCC) [31], colorectal cancer (CRC) [27], cervical cancer (CC) [28], lung cancer [32], and non-small cell lung cancer (NSCLC) [29]. In addition, it promotes the growth of the inner membrane of tissues in non-cancerous diseases such as endometriosis. Studies indicate that the growth of endometrial tissue outside the uterine cavity in vivo is inhibited by silencing *SNHG4* [30].

The aim of the study was to analyse of the level of *SNHG4* gene expression in patients with endometriosis and in the control group, to correlate of the obtained results with clinical–pathological data, and to determination of the significance of the obtained results in the context of the risk of endometriosis. 

## 2. Materials and Methods

### 2.1. Patients

The materials for genetic testing for gene expression in lncRNA *SNHG4* were tissue slices embedded in paraffin blocks from patients with endometriosis (n = 100) and from control (n = 100). Paraffin blocks came from the archives of the Department of Clinical Pathology of the Polish Mother’s Memorial Hospital Research Institute, Lodz, Poland. All samples have been characterized histologically. The study group consisted of 100 patients operated on in the Department of Gynecology, Oncological Gynaecology and Endometriosis Treatment of the Polish Mother’s Memorial Hospital Research Institute in 2015–2016 due to endometriosis, which was confirmed during the operation by intraoperative examination and in the final result of the histopathological examination. The control group consisted of 100 patients from the of Gynecology, Oncological Gynaecology and Endometriosis Treatment of the Polish Mother’s Memorial Hospital Research Institute, in whom a trial abrasion was performed due to uterine myomas. In this group, normal endometrium was found in the histopathological examination. These patients did not have endometriosis. The characters of the patients will be presented in the Table 1. The clinical stage of endometriosis has been determined according to rASRM (The Revised American Society for Reproductive Medicine classification of endometriosis, 1996) [33]. Formal consent (No. 88/2022) was obtained from the Bioethics Committee at the Institute of the Polish Mother’s Memorial Hospital in Lodz, Poland.

### 2.2. RNA Isolation

Total RNA was isolated using the reagent Trizol (Ambion, TX, 78744-1038 USA). Fragments of the test preparations were suspended in 500 μL of buffer containing 10 mM NaCl, 500 mM Tris (pH 7.6), 20 mM EDTA, 1% SDS and 500 μL/mL proteinase K. Incubation was carried out all night at 50 °C. RNA was isolated from the obtained lysates using TRI reagent (Sigma Aldrich, Darmstadt, Germany). The RNA isolation procedure was carried out in accordance with the manufacturer’s recommendations.

### 2.3. Spectrophotometric Analysis of Purity and RNA Concentration

The purity of the obtained RNA preparations was determined by the spectrophotometric method. The absorbance of the sample was measured twice at wavelengths of 260 nm and 280 nm (RNA purity criterion; value A260/A280 within 1.8–2.0). The concentration of RNA was determined at a wavelength of 260 nm. This value corresponded to the following relationship:1OD = 40 µg RNA/mL.

The mean values of purity and RNA concentration obtained for the test material and the correct one met the necessary criteria.

### 2.4. RT-PCR Methods

The reverse transcription reaction was analyzed using the TaKaRa RNA PCR Kit (AMV) ver 3.0 (Takara Bio INC, Kusatsu, Shiga 525-0058, Japan). We suspended 1 μg of total RNA (1 μg/μL) in a 20 μL mixture containing 1 μL of oligo dT-Adaptor primer, 2 μL of reaction buffer 10 x RT Buffer, 4 μL of MgCl2, 0.5 μL of RNaz inhibitor, 2 μL dNTP (10 mM), 1 μlamV XL reverse transcriptase, and 7.5 μL DEPC treated water. The reaction conditions were as follows: 10 min at 30 °C, then 30 min at 42 °C, 5 min at 95 °C, and 5 min at 5 °C. The obtained cDNA was stored at −20 °C.

The reaction mixture contained 0.5 μL cDNA, 0.5 μL 20x TaqMan^®^ Gene Expression Assays (Applied Biosystems, Lincoln Centre Dr, Foster City, CA, 94404-1128, USA), 5 μL TaqMan^®^ Universal PCR MasterMix (Applied Biosystems, Lincoln Centre Dr, Foster City, CA, 94404-1128, USA) containing TaqMan^®^ DNA polymerase, dNTP, reaction buffer, and 4 μL of water. The real-time PCR reaction was performed in the Mastercycler^®^ ep realplex (Eppendorf, Hamburg, Germany). The thermal conditions of the reaction were as follows: initial denaturation at 95 °C for 10 min, 50 cycles involving incubation for 15 s at 95 °C and 1 min at 60 °C. The following commercially available sets of probes and primers were used in the real-time PCR reaction (OriGene Technologies GmbH, Schillerstr.532052, Herford, Germany). GAPDH was used for internal reference. The primer sequences were SNHG4 forward primer: 5′-TTCAAGCGATTCTCGTGCC-3′, reverse primer: 5′-AAGATTGTCAAACCCTCCCTGT-3; GAPDH forward primer: 5′-ACAACTTTGGTATCGTGGAAGG-3′; reverse primer: 5′-GCCATCACGCCACAGTTTC-3′. The obtained Ct values were converted into the number of mRNA copies of the tested genes per 1000 copies of the GAPDH gene mRNA according to the following relationships:ΔC_t_ = C_ttest gene_ − C_treference gene_
L = 1000 × 2^−ΔCt^
where L is the number of mRNA copies of the test gene/1000 copies of the mRNA of the reference gene. The mean Ct values obtained for the cancer material and the normal one met the necessary criteria.

### 2.5. Statistical Analysis

Statistical analysis of the results was performed using the STATISTICA 11 program (StatSoft, Krakow, Poland). Due to the lack of normal distribution of the obtained results, the analysis of the significant differences in the level of gene expression at the mRNA level was carried out using non-parametric tests (Mann–Whitney U test, Kruskal–Wallis test). This was confirmed by the Shapiro–Wolf test. The correlation analysis between the variables was performed using the Spearman-R test. Statistical significance was found at *p* < 0.05.

## 3. Results

The results of *SNHG4* expression in endometriosis patients and in the control group are presented in Table 2. Statistically significantly higher *SNHG4* expression was observed in cases of endometriosis compared to controls (Figure 1).

*SNHG4* lncRNA sequence expression was also statistically analyzed for correlation with clinical–pathological data on age (Figure 2), body mass index (BMI) (Figure 3), parity (Table 3), spontaneous abortion (Figure 4), and rASRM. An association was observed between *SNHG4* gene expression and rASRM classification in the endometriosis group (Figure 5). The expression of SNHG4 in patients with stage 1–2 endometriosis was statistically significantly lower compared to patients with stage 3–4 endometriosis (Table 4). The high expression of *SNHG4* was significantly associated (*p* = 0.031) with advanced stages of endometriosis. All the other correlations mentioned above (age, BMI, parity, spontaneous abortion) turned out to be statistically insignificant.

## 4. Discussion

As presented in the introduction, *SNHG4* transcription plays an important role in key biological processes occurring in cancer and in non-cancerous diseases such as endometriosis. SNHG4 has been extensively studied for cancer. However, it remains mostly unexplored in the case of endometriosis.

In the present work, we presented an analysis of *SNHG4* expression in patients with endometriosis compared to the control group. We correlated our results with clinical and pathological data. According to the available literature, *SNHG4* expression has not yet been studied in endometriosis. Therefore, our research may provide new data on the importance of this transcript in endometriosis.

Endometriosis can be considered a mild metastatic disease and, due to the ability of endometrial tissue to infiltrate and form metastases and relapses such as tumors, it is very similar to cancer [34]. Epidemiological data suggest that endometriosis has malignant potential [35]. Recent studies on endometriosis have highlighted lncRNA *SNHG4*, whose impaired expression can be observed in many cancers [10]. High *SNHG4* expression is closely related to the clinical–pathological features and prognosis of some cancers; high expression of *SNHG4* represents a worse prognosis. *SNHG4* can therefore be used as a prognostic biomarker for these diseases [36,37,38,39]. Experimental studies have shown that *SNHG4* can promote cell proliferation, invasion, migration, and EMT (the epithelial–mesenchymal transition) and inhibit apoptosis [37,40].

*SNHG4* can act as a sponge for miRNAs [29,41]. An example is *SNHG4* miR-204-5p sponges; they increase the expression of RRM2 which causes the occurrence of gastric cancer. Similarly, the miR-148a-3p/c-Met axis and the miR-let-7e/KDM3A/p21 axis contribute to the formation and development of cervical cancer and NSCLC. In inhibiting the survival and development of cancer cells, targeted inhibition of *SNHG4* appears to be beneficial. Therefore, *SNHG4* can become a potential therapeutic target for various diseases. There are few reports of the role of *SNHG4* in non-cancerous diseases. These include, for example, acute cerebral infarction (ACI), neonatal pneumonia (NP), and diabetic retinopathy (DR) [23,24,42], as well as endometriosis [30]. Future research should focus on these diseases.

The present work presents an analysis of lncRNA expression in patients with endometriosis compared to the control group. The aim of the study was to discern possible correlation of *SNHG4* with the risk of the above-mentioned disease. Our statistical analysis showed a significant correlation of *SNHG4* expression levels with endometriosis. In cases of disease, *SNHG4* showed a higher level of expression than in controls. *SNHG4* has been shown to be significant in terms of its correlation with the pathological data of cases.

The expression level of *SNHG4* correlated with the stage of endometriosis. The results presented in this paper regarding the analysis of *SNHG4* reveal the existence of certain relationships between its expression and endometriosis. It should be noted, however, that the present research covered a small population (100 patients and 100 control) and requires further work to be carried out on much larger groups of respondents. The study groups we use may simply be quantitatively unsatisfactory to draw the right conclusions. In addition, the cases and controls are not homogenous; they differ in age to some extent, which can falsify the results. Moreover, the cases are disease-free women; they were all treated surgically for a mild gynecological condition: symptomatic uterine fibroids. There are data that some lncRNAs (e.g., H19 lncRNA) may show correlation with uterine fibroid tissue [43]. We want to point out that, in our study, genetic tests were carried out strictly on selected endometrial tissues and not on uterine fibroids. Therefore, taking into consideration the preliminary data of our pilot study, we believe that further research in this field is needed.

## 5. Conclusions

According to our pilot study, elevated SNHG4 expression levels may be associated with endometriosis. The state of knowledge we currently have concerning SNHG4 in endometriosis remains limited. Therefore, further research is necessary and justified in order to further explore this topic.

## Figures and Tables

**Figure 1 genes-14-00152-f001:**
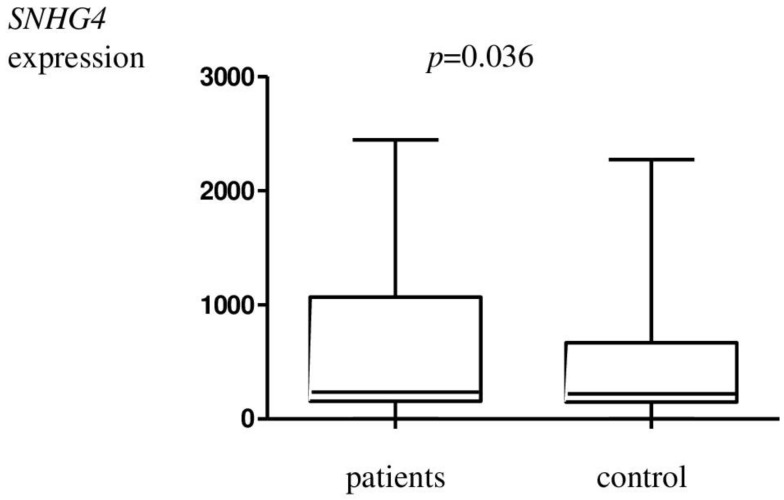
Relative expression of the *SNHG4* gene in the test group with respect to the control material.

**Figure 2 genes-14-00152-f002:**
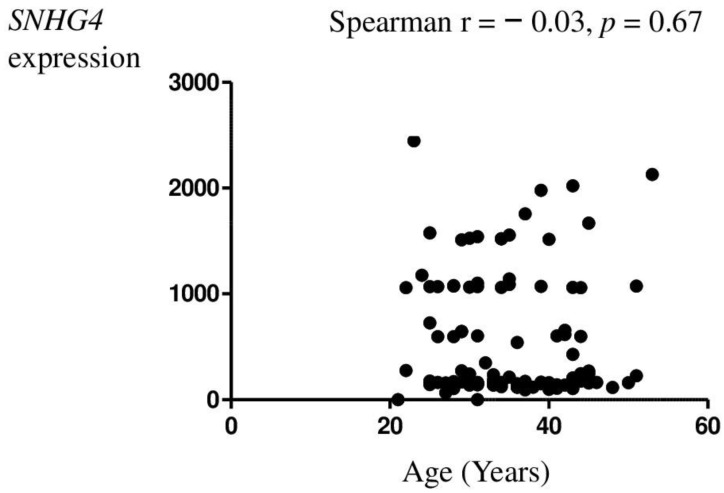
Relationship between patient age and relative expression of the *SNHG4* gene.

**Figure 3 genes-14-00152-f003:**
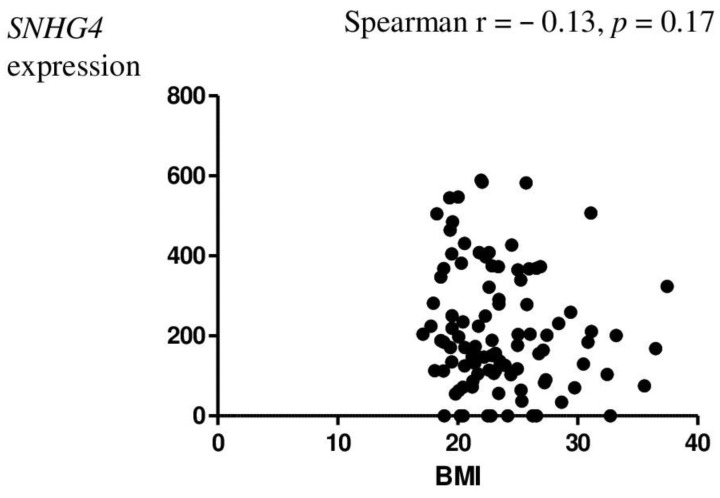
Correlation between patient BMI and relative expression of the *SNHG4* gene.

**Figure 4 genes-14-00152-f004:**
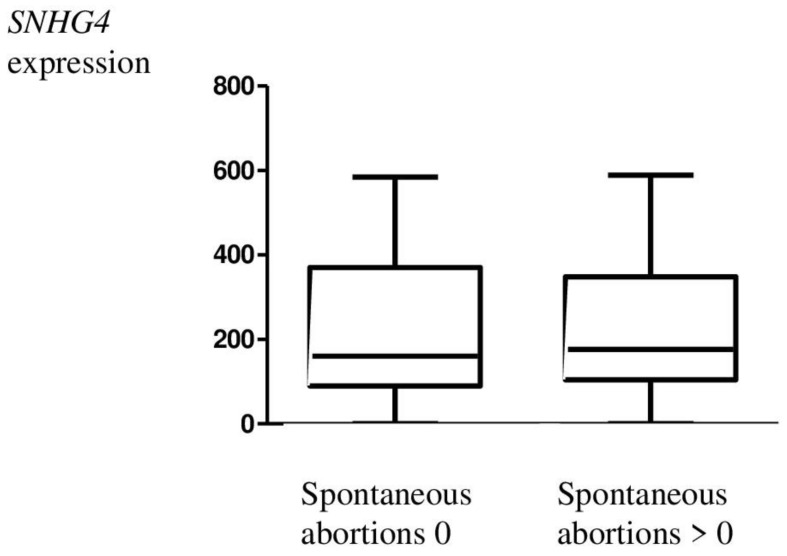
Relative expression of the *SNHG4* gene in relation to the number of spontaneous abortions (Mann–Whitney test; *p* = 0.68).

**Figure 5 genes-14-00152-f005:**
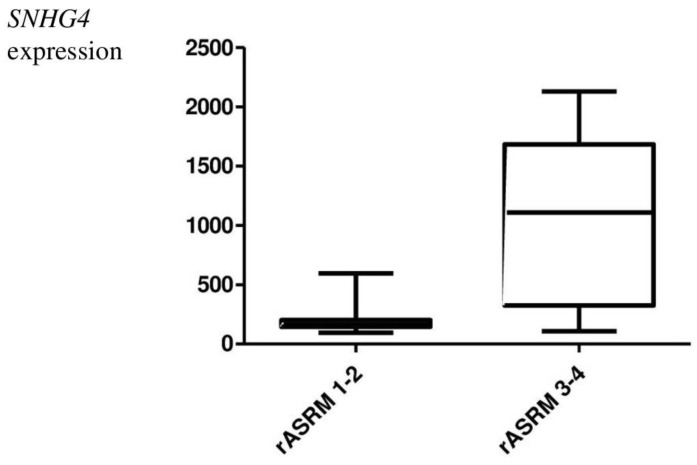
Relative expression of the *SNHG4* gene in endometriosis patients with respect to the rASRM classification (Mann–Whitney test; *p* = 0.031).

**Table 1 genes-14-00152-t001:** Clinical–pathological characteristics of the study groups.

Patients (n = 100)		Control (n = 100)	
Age (Range) 21–53 Years Age (Mean) 34.89 ± 7.49		Age (Range) 26–67 Years Age (Mean) 47.93 ± 6.01	
BMI (kg/m^2^)	The number (%)	BMI (kg/m^2^)	The number (%)
<25	65 (65%)	<25	55 (55%)
25 ≤ BMI < 30	25 (25%)	25 ≤ BMI < 30	24 (24%)
≥30	10 (10%)	≥30	21 (21%)
Parity	The number (%)	Parity	The number (%)
yes	45 (45%)	yes	50 (50%)
−1	20 (20%)	−1	25 (25%)
−2 and more	24 (24%)	−2 and more	20 (20%)
no	11 (11%)	no	5 (5%)
*Spontaneous abortions*	The number (%)	*Spontaneous abortions*	The number (%)
yes (spontaneous)	6 (6%)	yes (spontaneous)	10 (10%)
no	94 (94%)	no	90 (90%)
Clinical stage	The number (%)		
I	24 (24%)		
II	27 (27%)		
III	18 (18%)		
IV	31 (31%)		

**Table 2 genes-14-00152-t002:** Expression of *SNHG4* in endometriosis patients and control.

SNHG4	Test Material	Median	Percentile 25	Percentile 75	*p* ^a^
	endometriosis	203.92	170.48	1011.22	<0.050
	control	201.878	168.74	768.64	

^a^ Mann-Whitney U test.

**Table 3 genes-14-00152-t003:** The expression of the *SNHG4* gene in relation to the number of parity (Mann–Whitney test; *p* > 0.05).

	Mann-Whitney Test	
	*p* = 0.78	
	Parity = 0	Parity > 0
number of preparations	38	62
average ± SD	214.3± 186.0	206.1 ± 154.3

**Table 4 genes-14-00152-t004:** The expression of the *SNHG4* gene in relation to the rASRM (Mann–Whitney test; *p* < 0.05).

	Mann-Whitney Test	
	*p* = 0.031	
	rASRM 1–2	rASRM 3–4
number of preparations	100	100
min.	126.67	128.10
max	670.11	2180.12
average ± SD	398.3 ± 186.0	1154.1 ± 154.3

## Data Availability

All data and materials, as well as software, confirm the published claims and comply with field standards.

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
