# Peer review of "Long Non-Coding RNA SNHG4 Expression in Women with Endometriosis: A Pilot Study"

_genes, 2023, doi:10.3390/genes14010152_

Round 1
Reviewer 1 Report
Endometriosis is a debilitating disease affecting women from adolescence until the perimenopausal period. The means to discover at the earliest each woman who suffers from this affliction are still under research, as, for now, the tests we have are not reliable laboratory biomarkers.
The current study analyzes the potential of a new biomarker as a testing tool to diagnose endometriosis. The idea is valuable, but I believe there are issues to be modified before considering this material for publication.
Lines 33-34 Please specify that this prevalence refers to the general women population (compared with the infertile women population and those with chronic pelvic pain).
Lines 72 -80 Is the mouse model important for the current hypothesis? It is linked mainly to cancer cell development; thus, it doesn’t quite matter for the current topic.
Lines 104 – 107 The control group did not consist of healthy subjects. It is not appropriate to assume that some endometrial changes could not be associated with fibroids.
Table 1 The maximum age range for the Control group was 67 years – how thick should the endometrium be in the menopause period? Perhaps a few patients should not have been included in the control group.
Author Response
Thank you for your review.
I would like to kindly ask you to reconsider the publication of our revised paper:
Long non-coding RNA SNHG4 expression in women with endometriosis
I hereby provide responses to the reviewers and list the changes that have been made in the revised version of our paper.
Rev 1
Endometriosis is a debilitating disease affecting women from adolescence until the perimenopausal period. The means to discover at the earliest each woman who suffers from this affliction are still under research, as, for now, the tests we have are not reliable laboratory biomarkers.
The current study analyzes the potential of a new biomarker as a testing tool to diagnose endometriosis. The idea is valuable, but I believe there are issues to be modified before considering this material for publication.
1Lines 33-34 Please specify that this prevalence refers to the general women population (compared with the infertile women population and those with chronic pelvic pain).
Ad1 The sentence has been corrected.
2 Lines 72 -80 Is the mouse model important for the current hypothesis? It is linked mainly to cancer cell development; thus, it doesn’t quite matter for the current topic.
Ad2 We removed the paragraph on mouse research.We agree that the mouse model is not necessary for the hypothesis, but we wanted to introduce in the introduction, based on the latest literature, how the research on the function and role of SNHG4 lncRNA has proceeded. We believe this paragraph may have been relevant for readers to fully understand the role of SNHG4 in a variety of diseases, including endometriosis.
3Lines 104 – 107 The control group did not consist of healthy subjects. It is not appropriate to assume that some endometrial changes could not be associated with fibroids.
Ad3 We agree, the control cases are not a group of women free from the disease, but all of them were treated surgically for a mild gynecological disease - symptomatic uterine fibroids. Some reports suggest that some lncRNA sequences (e.g. lncRNA H19) may correlate with uterine fibroids as such [1], but in our study genetic testing was performed strictly on selected endometrial tissues and not on uterine fibroids. The traditional “manual macrodissection” method for dissecting formalin-fixed paraffin-embedded (FFPE) tissue slides was used.The method was carried out by pathologists from the Department of Pathology Polish Mothers Memorial Hospital Research Institute.
- Cao T, Jiang Y, Wang Z, Zhang N, Al-Hendy A, Mamillapalli R, et al. H19 lncRNA identified as a master regulator of genes that drive uterine leiomyomas. Oncogene. 2019;38(27):5356–66.
4Table 1 The maximum age range for the Control group was 67 years – how thick should the endometrium be in the menopause period? Perhaps a few patients should not have been included in the control group.
Ad4 Thanks for your comments. There are some limitations of our study that need to be mentioned and considered with criticism. The results presented in this paper on the analysis of lncRNA gene reveal the existence of certain relationships of their expression with endometriosis. It should be noted, however, that the presented research covered a small population (100 patients and 100 control) and requires further work carried out on much larger groups of respondents. The study groups we use may simply be quantitatively unsatisfactory to draw the right conclusions. Second, the cases and controls are not exactly homogeneous – to some extent they differ in age, which can falsify the results.We treat this research as a pilot. We want to continue the research and, of course, select more homogeneous groups. With all of the above findings in mind, and realizing the limitations of our study, we dare say that this research has shed new light on lncRNAs in endometriosis and contributes to a growing – but still unclear – knowledge of these non-coding sequences in endometriosis.
I hope you find our revised Manuscript satisfying so that it can meet the criteria of publication in your Journal.
Looking forward to hearing from you,
Yours sincerely,
Beata Smolarz
Reviewer 2 Report
Ms. Ref. No.: genes-1970393
Title: Long non-coding RNA SNHG4 expression in women with endometriosis
This study aim to understand the correlation between the expression of lncRNA small nucleolar RNA host gene 4 (SNHG4) and the risk of endometriosis. The introduction is informative. The authors introduce the relevant data for the objectives of the study.
RNA from FFPE from 100 patients with endometriosis were subjected to the RT-PCR. Their findings showed that elevated SNHG4 expression levels may be associated with endometriosis. Although the experiments are reasonably well designed, as the author mentioned that elevated SNHG4 expression levels in patients with endometriosis, the main limitation of this study is no functional validation of SNHG4 to demonstrate its role in the pathomechnism of endometriosis.
Minor comments:
(1). Please review and edit the English in your paper - prior to submission.
(2). Abbreviations should be defined at first mention and used consistently thereafter
(3). Line 96: Authors should describe how they obtained the FFPE tissues. Macrodissection or microdissection?
(4). I think Figure 2-5 could be expressed as supplementary data.
(5). The association between SNHG4 and rASRM classification should be discussed in detailed.
(6). Line 240: How to make a conclusion that elevated SNHG4 expression levels have been linked to endometriosis without any functional assay?
Author Response
Rev2
Thank you for your review.
I would like to kindly ask you to reconsider the publication of our revised paper:
Long non-coding RNASNHG4 expression in women with endometriosis
I hereby provide responses to the reviewers and list the changes that have been made in the revised version of our paper.
This study aim to understand the correlation between the expression of lncRNA small nucleolar RNA host gene 4 (SNHG4) and the risk of endometriosis. The introduction is informative. The authors introduce the relevant data for the objectives of the study.
RNA from FFPE from 100 patients with endometriosis were subjected to the RT-PCR. Their findings showed that elevated SNHG4 expression levels may be associated with endometriosis.Although the experiments are reasonably well designed, as the author mentioned that elevated SNHG4 expression levels in patients with endometriosis, the main limitation of this study is no functional validation of SNHG4 to demonstrate its role in the pathomechnism of endometriosis.
Minor comments:
(1). Please review and edit the English in your paper - prior to submission.
We tried to edit the English, other reviewers had no comments on the language.
(2). Abbreviations should be defined at first mention and used consistently thereafter
Ad2 We corrected the Abbreviations
(3).Line 96: Authors should describe how they obtained the FFPE tissues. Macrodissection or microdissection?
Ad3 The traditional “manual macrodissection” method for dissecting formalin-fixed paraffin-embedded (FFPE) tissue slides was used.The method was carried out by pathologists from the Department of Pathology Polish Mothers Memorial Hospital Research Institute.
(4). I think Figure 2-5 could be expressed as supplementary data.
Ad4 Figure have been included as supplementary data.
(5). The association between SNHG4 and rASRM classification should be discussed in detailed.
Ad5 The data is described in the results section.
(6). Line 240: How to make a conclusion that elevated SNHG4 expression levels have been linked to endometriosis without any functional assay?
Ad6 Thanks for your comments. We rearranged conclusion. We treat the research as pilot.
There are some limitations of our study that need to be mentioned and considered with criticism. The dominant shortcoming of our analysis are numbers: a genetic research study on 100 Cases and 100 Controls (200 assays in total) makes an experienced researcher draw final conclusions with care and skepticism. Especially in genetics, our groups may be simply quantitatively unsatisfactory to conduct a fair judgement. We treat the conclusion that increased levels of SNHG4 expression was associated with endometriosis very carefully. We intend to continue the research as the results are promising. In subsequent works, we intend to increase the study group and include functional tests. However, such research requires funding. If we only obtain a grant, we will certainly take into account this important suggestion regarding functional tests.
Having in mind all the abovementioned findings and aware of the restrictions of our study, we dare to claim that this research has shed some new light on lncRNA in endometriosis and contributes to the growing – but still fragmentary – body of knowledge on these non-coding sequences in endometriosis.
I hope you find our revised Manuscript satisfying so that it can meet the criteria of publication in your Journal.
Looking forward to hearing from you,
Yours sincerely,
Beata Smolarz
Round 2
Reviewer 1 Report
Dear Authors,
I appreciate the effort made to improve the submitted material. Still, there are crucial issues that were not resolved and can not be resolved. Whenever a theory is under debate, the two study groups which sustain the research should have similar characteristics. In the first lines from the Introduction section (35-37) endometriosis is defined as a disease of young women, sustaining that age is important. Table 1 from the Materials and Methods Section describes a mean age of 34,89 years in the study group and 47,93 in the control group 9 (meaning that most of the cases included are postmenopausal women). Even if a Limitations subsection would be included in the Discussions section, I strongly consider comparing such extent different age groups with this pathology would deliver biased results.
Author Response
Thank you very much for your valuable comments. We can only add in defense of the work that endometriosis can affect 2-5% of postmenopausal women. We have added a fragment in the Introduction section. In our study group, there were about 2% of postmenopausal patients (patients range 21-53, controls 26-67). We agree that the groups could have been more homogeneous, but we ask for your understanding. We have added the limitation of the study. As we have noted, we treat the results carefully and as a pilot study. Given the preliminary data from our pilot study and the limitations of our study, we believe that more research is needed in this regard.Having in mind all the abovementioned findings and aware of the restrictions of our study, we dare to claim that this research has shed some new light on lncRNA in endometriosis and contributes to the growing – but still fragmentary – body of knowledge on these non-coding sequences in endometriosis.
Once again, please respond positively to our research.
Best Regards
Beata Smolarz
Reviewer 2 Report
The revised version of the manuscript by Szaflik T et al. represents a considerable improvement over the initial version. The work is now suitable for publication
In Biomedicines.
Author Response
Thank you for your positive review
Best Regards
Beata Smolarz